# Effects of Exercise Type on Muscle Strength and Body Composition in Men and Women: A Systematic Review and Meta-Analysis

**DOI:** 10.3390/medicina60071186

**Published:** 2024-07-22

**Authors:** Ki-Woong Noh, Eui-Kyoung Seo, Sok Park

**Affiliations:** 1Institute of Sports Medicine & Science, Kwangwoon University, Seoul 01897, Republic of Korea; shrldn123@kw.ac.kr; 2Division of Law, Kwangwoon University, Seoul 01897, Republic of Korea; seo0903@kw.ac.kr

**Keywords:** endurance training, resistance training, concurrent training, muscle strength, body composition, sex characteristics

## Abstract

*Background and Objectives:* There are typical differences in body composition and distribution of muscle fiber types between women and men. However, research investigating the effects of exercise based on sex differences is limited, and studies examining sex differences in physiological adaptations according to exercise type are scarce. We aimed to compare the effects of exercise types on muscle strength and body composition in men and women through a meta-analysis. *Materials and Methods:* A systematic literature search was conducted using the PubMed/Medline, Web of Science, CINAHL, and EBSCO databases. Keywords included “endurance training”, “resistance training”, “concurrent training”, “muscle strength”, “body composition”, “sex characteristics”, and “men and women”. The standardized mean difference (SMD) was presented separately for men and women based on the pre- and post-intervention values for each exercise type. *Results:* Concurrent training showed the greatest effect on the increase in leg press muscle strength in men, and resistance training showed the greatest effect in women. Concurrent training showed the greatest effect size in both men and women in increasing bench press muscle strength. Resistance training and concurrent training showed a small effect size on lean mass reduction in both men and women. Endurance training and concurrent training significantly reduced fat mass in men. However, no significant changes in fat mass were observed in any exercise type among women. *Conclusions:* Concurrent training is the most efficient type of exercise for men, as it is effective in increasing upper- and lower-body muscle strength, increasing lean mass, and reducing fat mass. Resistance training is most effective in increasing muscle strength in females, whereas endurance training is most effective in reducing fat mass. However, it is difficult to corroborate these results because of the lack of study samples included in the analysis and the differences in exercise methods, participant age, and exercise duration.

## 1. Introduction

Aging and lack of physical activity are major global public health issues [1]. These problems cause sarcopenia and obesity and affect the occurrence of several related diseases. Sarcopenia affects tens of millions of adults worldwide, with a notably high prevalence among the elderly [2,3]. According to the World Health Organization (WHO), only 27–44% of elderly individuals in the United States meet the recommendations for physical activity. Problems such as muscle strength and mass loss are observed not only in the elderly but also in numerous young adults due to decreased physical activity and overnutrition and are also associated with an increased prevalence of sarcopenic obesity [4,5]. According to the WHO, 43% of adults were overweight, and more than 1 billion people globally were living with obesity in 2022, with adult obesity more than doubling since 1990 [6]. Physical activity plays the most important role in preventing and treating problems such as sarcopenia, sarcopenic obesity, and obesity [7,8]. However, owing to an accelerated aging population, urbanized lifestyles, sedentary habits, and excessive nutritional intake, the time and environment for proper physical activity have become scarce [5]. Therefore, it is necessary to provide effective exercise methods tailored to goals and circumstances to prevent and address major public health issues, such as sarcopenia and obesity. RT strongly stimulates muscle metabolism and protein synthesis, promoting an increase in muscle strength and mass [9]. Physiological adaptations to RT play crucial roles in the management of obesity, muscle atrophy, and physical frailty [10]. However, physical adaptation altered by RT intervention is highly variable and depends on personal characteristics such as sex, race, and age [11]. Therefore, understanding bodily adaptations to ET and RT, considering personal characteristics and physiological differences, can provide important information for exercise programs aimed at optimally improving body function and composition.

ET increases capillary density and induces an increase in the maximal oxygen consumption rate [12]. In contrast, RT hypertrophies the muscle fibers and simultaneously decreases capillary density [13]. Furthermore, differences in the types of muscle fibers that increase in response to ET and RT interventions are apparent. ET induces a greater increase in type I muscle fibers than RT, with less pronounced augmentation of type II muscle fibers [14]. Additionally, research findings suggest that high-intensity ET may decrease muscle fiber cross-sectional area [15]. However, there are differences in body composition changes following RT and ET interventions depending on the exercise intensity, frequency, and duration [16]. Concurrent training (CT), a combination of RT and ET, results in greater physical ability improvement and body composition changes than RT or ET alone when exercise methods are appropriate [17,18]. However, excessive exercise intensity or volume and inappropriate exercise programming in CT can hinder or suppress neuromuscular adaptations and reduce endurance [19]. Given that the exercise type and individual characteristics significantly influence post-exercise physiological responses, comprehensive research is required to propose appropriate exercise methods that account for these factors.

There are typical differences in the body composition and distribution of muscle fiber types between women and men. Although men typically have a greater absolute muscle mass than women, the acceleration of muscle loss due to aging is faster in men than in women [20]. Previous studies suggest that sex differences in muscle mass and strength might influence the prevalence of sarcopenia [21,22]. Physical adaptation according to exercise intervention depends on sex differences such as hormones, inflammatory reactions, fatigue, muscle fiber types, and energy metabolism [23]. Therefore, sex-specific interventions must be considered when developing exercise programs to minimize muscle loss, increase physical function, and optimize body composition [22,24]. ET, RT, and CT have received increasing attention because of their efficacy in improving physical function and body composition. However, research investigating the effects of exercise based on sex differences is limited, and studies examining sex differences in physiological adaptations according to exercise type are scarce.

Therefore, we aimed to compare the effects of different types of exercise on muscle strength and body composition between men and women through a meta-analysis to provide valuable information for the design of practical exercise programs tailored to sex and for further research on differences in exercise types.

## 2. Materials and Methods

This review was conducted in accordance with the Preferred Reporting Items for Systematic Reviews and Meta-Analyses guidelines [25] and the National Evidence-Based Healthcare Collaborating Agency guidance for systematic reviews and meta-analyses [26]. This study was registered in PROSPERO (ID: CRD42024549835).

### 2.1. Search Strategy

We performed literature searches of the PubMed/Medline, Web of Science, and CINAHL databases from their inception to 30 May 2024. The EBSCO database was used for extensive data searches. The same search term was used in all databases, and searches were conducted using the Medical Subject Headings for keywords such as “endurance training”, “resistance training”, “concurrent training”, “muscle strength”, “body composition”, and “sex characteristics”. Specifically, we searched for the following combinations of keywords: [resistance training, endurance training, concurrent training] [muscle strength, body composition] and sex. The language of the included studies was limited to English.

### 2.2. Inclusion Criteria

We conducted a systematic review to identify randomized controlled trials (RCTs) investigating the effects of exercise type on muscle strength and body composition in men and women. In this study, for individual studies of men or women, only those that included ET, RT, and CT groups were included in the analysis. For studies that directly compared the outcomes between men and women, those that included one or more ET, RT, or CT groups were included in the analysis. The PICOS model was used to determine the criteria for the studies to be included in the analysis [26,27]. The specific criteria of our search were as follows: P (population): men and women; I (intervention): ET, RT, and CT; C (comparator): the pre- and post-values within each group were compared; O (outcome): mean ± standard deviation data (muscle strength, lean body mass, fat mass); S (study design): This systematic review included only RCT studies.

### 2.3. Exclusion Criteria

The following studies were excluded: (1) research on animals, children, adolescents, obese people, and patients and studies in which male and female data were not distinguished; (2) nonrandomized controlled trials; (3) studies that included interventions other than the study’s purpose; (4) studies that did not include muscle strength, lean body mass, or fat mass as outcome indicators; (5) cohort studies, quasi-experimental studies, qualitative studies, meta-analyses, and reviews; and (6) studies not published in English.

### 2.4. Quality Assessment and Data Extraction

Three authors (K.W.N., S.P., and E.K.S.) independently evaluated the methodological quality of the included studies by using the Cochrane risk of bias tool for RCTs [26,28]. The items in the tool were divided into seven specific domains: (1) random sequence generation (selection bias); (2) allocation concealment (selection bias); (3) blinding of participants and personnel (performance bias); (4) blinding of outcome assessment (detection bias); (5) incomplete outcome data (attrition bias); (6) selective reporting (reporting bias); and (7) other sources of bias.

### 2.5. Statistical Analysis

The descriptive data of the participant characteristics are presented as means. The meta-analysis was conducted using STATA/SE 18 version (Stata Corp, Station, TX, USA). The standardized mean difference (SMD), number of participants, and standard error in the SMD for each study were used to quantify the changes in the dependent variables when comparing intragroup pre- and post-intervention. The SMD for each study was calculated using the Hedges’ g random-effects model (I^2^ ≥ 50%) or Hedges’ g fixed-effects model (I^2^ < 50%) [28]. Considering the methodological heterogeneity across the included studies, a random- or fixed-effects model was used to quantify the pooled SMD of the included studies [29]: small, SMD = 0.2; medium, SMD = 0.5; and large, SMD = 0.8.

Statistical heterogeneity was assessed using Cochran’s Q test and I^2^ statistics. When heterogeneity was observed between studies, a random-effects model was applied; when homogeneity was observed, a fixed-effects model was applied. Furthermore, the possibility of publication bias was estimated by visually inspecting the funnel plot using a contour-enhanced funnel plot when at least 10 studies were included in the meta-analysis [28].

## 3. Results

### 3.1. Literature Search Results

A systematic literature search yielded 7259 articles that were included in the analysis. A total of 3815 duplicate articles were excluded from 7259 articles searched. A review of the titles and abstracts of the articles led to the exclusion of 3092 studies that were deemed unrelated. Sixty-one studies that could not be secured during the download and collection phases to confirm the full text were excluded. After a full-text review of the 291 articles, 266 studies were excluded for the following reasons: A total of 65 had incomplete data, 159 did not meet the inclusion criteria, and 42 were not RCTs. After the systematic literature search, 25 articles were included (Figure 1).

### 3.2. Quality Assessment

Each study was judged as having a low, high, or unclear risk of bias (Figure 2 and Figure 3). The studies included in our meta-analysis were assessed as having a high or unclear risk of random sequence generation (6/25), allocation concealment (12/25), blinding of participants and personnel (9/25), blinding of outcome assessment (23/25), incomplete outcome data (6/25), selective reporting (14/25), or other sources of bias (15/25).

### 3.3. Study Characteristics

Table 1 shows the characteristics of the 25 studies included in the meta-analysis [16,18,30,31,32,33,34,35,36,37,38,39,40,41,42,43,44,45,46,47,48,49,50,51,52]. A total of 1102 participants were included, with an average age of 38.98 years. Among these, 747 men had an average age of 38.52 years, and 355 women had an average age of 39.95 years. Thirteen studies investigated the effects of ET, RT, and CT on men. Four studies investigated the effects of ET, RT, and CT on women. In addition, eight studies investigated the effects of exercise type in men and women. The average exercise period in 21 studies involving men is 14 weeks, and the average exercise period in 12 studies involving women is 16.9 weeks.

### 3.4. Effect of Exercise Type on Leg Press Muscle Strength in Men and Women

#### 3.4.1. Effect of CT

Figure 4 presents the results of a meta-analysis of the effect of CT interventions on leg press muscle strength in men and women. The pooled SMD, which analyzed the results of the men using the Hedges’ g random-effects model, was 2.33 (95% CIs, 1.40 to 3.25; *p* = 0.000; MSMD, large), indicating a significant increase in leg press muscle strength in men after the CT intervention. The I^2^ statistic indicated heterogeneity (I^2^, 83.8%, *p* = 0.000). The pooled SMD, analyzing the results of the women group, was 3.29 (95% CIs, 1.41 to 5.17; *p* = 0.001; MSMD, large), indicating a significant increase in leg press strength after CT intervention. The I^2^ statistic indicated statistical heterogeneity (I^2^, 83.8%, *p* = 0.000). The CT showed a large effect size for leg press strength in both men and women.

#### 3.4.2. Effect of RT

Figure 5 presents the results of a meta-analysis of the effect of RT intervention on leg press muscle strength in men and women. The pooled SMD, which analyzed the results of the men using the Hedges’ g random-effects model, was 2.21 (95% CIs, 0.23 to 3.18; *p* = 0.000; MSMD, large), indicating a significant increase in leg press muscle strength in men after RT intervention. The I^2^ statistic indicated statistical heterogeneity (I^2^, 87.3%, *p* = 0.000). The pooled SMD, analyzing the results of the women group, was 3.38 (95% CIs, 1.04 to 5.71; *p* = 0.005; MSMD, large), indicating a significant increase in leg press strength after RT intervention. The I^2^ statistic indicated statistical heterogeneity (I^2^, 89.7%, *p* = 0.000). RT showed a large effect size in both the male and female groups for leg press strength.

#### 3.4.3. Effect of ET

Figure 6 presents the results of a meta-analysis of the effect of ET intervention on leg press muscle strength in men and women. The pooled SMD, which analyzed the results of the men using the Hedges’ g random-effects model, was 1.08 (95% CIs, 0.37 to 1.79; *p* = 0.003; MSMD, large), indicating a significant increase in leg press muscle strength in men after ET intervention. The I^2^ statistic indicated statistical heterogeneity (I^2^, 75.2%, *p* = 0.000). The pooled SMD, analyzing the results of the women group, was 1.72 (95% CIs, −0.38 to 3.81, *p* = 0.108; MSMD, large), indicating a significant increase in leg press strength after ET intervention. The I^2^ statistic indicated statistical heterogeneity (I^2^, 86.4%, *p* = 0.001). ET showed a large effect size in both the male and female groups on leg press strength.

### 3.5. Effect of Exercise Type on Bench Press Muscle Strength in Men and Women

#### 3.5.1. Effect of CT

Figure 7 presents the results of a meta-analysis of the effect of CT intervention on bench press muscle strength in men and women. The pooled SMD, which analyzed the results of the men using the Hedges’ g random-effects model, was 2.25 (95% CIs, 0.77 to 3.74; *p* = 0.003; MSMD, large), indicating a significant increase in bench press. The pooled SMD, analyzing the results of the women group, was 2.89 (95% CIs, 0.25 to 5.54; *p* = 0.032; MSMD, large), indicating a significant increase in bench press strength after CT intervention. The I^2^ statistics indicated statistical heterogeneity (I^2^, 88.5%, *p* = 0.003). CT showed a large effect size in both the men and women groups in bench press strength.

#### 3.5.2. Effect of RT

Figure 8 presents the results of a meta-analysis of the effect of RT intervention on bench press muscle strength in men and women. The pooled SMD, which analyzed the results of the men using the Hedges’ g random-effects model, was 1.74 (95% CIs, 0.94 to 2.55; *p* = 0.000; MSMD, large), indicating a significant increase in bench press muscle strength in men after RT intervention. The I^2^ statistic indicated statistical heterogeneity (I^2^, 87.3%, *p* = 0.000). The pooled SMD, analyzing the results of the women group, was 1.97 (95% CIs, 0.90 to 3.03; *p* = 0.000; MSMD, large), indicating a significant increase in bench press strength after RT intervention. The I^2^ statistic indicated statistical heterogeneity (I^2^, 89.6%, *p* = 0.000). RT showed a large effect size in both the male and female groups in terms of bench press strength.

#### 3.5.3. Effect of ET

Figure 9 presents the results of the meta-analysis of the effect of ET intervention on bench press muscle strength in men and women. The pooled SMD, which analyzed the results of the men using the Hedges’ g random-effects model, was 0.59 (95% CIs, −0.14 to 1.32; *p* = 0.115; MSMD, medium), indicating a significant increase in bench press muscle strength in men after ET intervention. The I^2^ statistic indicated statistical heterogeneity (I^2^, 68.5%, *p* = 0.013). The pooled SMD, which analyzed the results of the men using the Hedges’ fixed-effects model, was 0.28 (95% CIs, −0.33 to 0.89; *p* = 0.364; MSMD, small; I^2^, 0%, *p* = 0.603), and ET showed a small effect size on increasing the bench press muscle strength of women. After the ET intervention, the male group showed a moderate effect on the increase in bench press 1 RM, which increased significantly. However, the ET intervention in increasing bench press 1 RM in the female group showed a small effect size and was found to be insignificant.

### 3.6. Effect of Exercise Type on Lean Body Mass in Men and Women

#### 3.6.1. Effect of CT

Figure 10 presents the results of a meta-analysis of the effect of CT intervention on lean body mass in men and women. The pooled SMD, which analyzed the results of the men using the Hedges’ g fixed-effects model, was 0.22 (95% CIs, −0.04 to 0.48; *p* = 0.091; MSMD, small), and CT intervention showed a small effect size on increasing the lean body mass in men. The I^2^ statistic showed no statistical heterogeneity (I^2^, 0%, *p* = 0.993). The pooled SMD, which analyzed the results of the women group, was 0.24 (95% CIs, −0.06 to 0.55; *p* = 0.121; MSMD, small), and ET showed a small effect size on increasing the lean body mass of women. The I^2^ statistic showed no statistical heterogeneity (I^2^, 0%, *p* = 1.000). CT showed a small effect on lean body mass in both men and women.

#### 3.6.2. Effect of RT

Figure 11 presents the results of a meta-analysis of the effect of RT intervention on lean body mass in men and women. The pooled SMD, which analyzed the results of the men using the Hedges’ g fixed-effects model, was 0.21 (95% CIs, −0.01 to 0.43; *p* = 0.056; MSMD, small), and RT intervention showed a small effect size on increasing the lean body mass in men. The I^2^ statistic showed no statistical heterogeneity (I^2^, 0%, *p* = 0.985). The pooled SMD, which analyzed the results of the women’s group, was 0.22 (95% CIs, −0.03 to 0.74; *p* = 0.000; MSMD, small), and RT showed a small effect size on increasing the lean body mass of women. The I^2^ statistic showed no statistical heterogeneity (I^2^, 0%, *p* = 0.794). RT showed a small effect on lean body mass in both men and women.

#### 3.6.3. Effect of ET

Figure 12 presents the results of a meta-analysis of the effect of ET intervention on lean body mass in men and women. The pooled SMD, which analyzed the results of the men using the Hedges’ g fixed-effects model, was −0.12 (95% CIs, −0.45 to 0.20; *p* = 0.465; MSMD, trivial), and ET intervention showed a very small effect size on lean body mass reduction in men. The I^2^ statistic showed no statistical heterogeneity (I^2^, 0%, *p* = 0.997). The pooled SMD, which analyzed the results of the women’s group, was 0.10 (95% CIs, −0.29 to 0.50; *p* = 0.607; MSMD, trivial), and ET showed a very small effect size on increasing the lean body mass of women. The I^2^ statistic showed no statistical heterogeneity (I^2^, 0%, *p* = 0.978). ET had a small effect on lean body mass in both men and women. ET intervention had a marginal and insignificant effect on the lean body mass of men and women.

### 3.7. Effect of Exercise Type on Fat Mass in Men and Women

#### 3.7.1. Effect of CT

Figure 13 presents the results of the meta-analysis of the effects of CT intervention on fat mass in men and women. The pooled SMD, which analyzed the results of the men using the Hedges’ g fixed-effects model, was −0.28 (95% CIs, −0.53 to −0.03; *p* = 0.029; MSMD, small), and CT intervention showed a small effect size on decreasing the fat mass in men. The I^2^ statistic showed no statistical heterogeneity (I^2^, 0%, *p* = 0.619). The pooled SMD, which analyzed the results of the women’s group, was −0.15 (95% CIs, −0.48 to 0.18; *p* = 0.000; MSMD, trivial), and CT showed a marginal effect size on decreasing the fat mass of women. The I^2^ statistic showed no statistical heterogeneity (I^2^, 0%, *p* = 0.978). CT showed a small effect size in fat mass reduction in men and a significant decrease; whereas, in women, CT showed a very small effect size in fat mass reduction, and no significant decrease was observed.

#### 3.7.2. Effect of RT

Figure 14 presents the results of a meta-analysis of the effect of RT intervention on lean body mass in men and women. The pooled SMD, which analyzed the results of the men using the Hedges’ g fixed-effects model, was −0.16 (95% CIs, −0.38 to 0.05; *p* = 0.129; MSMD, trivial), and RT intervention showed a marginal effect size on decreasing fat mass in men. The I^2^ statistic showed no statistical heterogeneity (I^2^, 0%, *p* = 0.942). The pooled SMD, which analyzed the results of the women’s group, was −0.17 (95% CIs, −0.42 to 0.08; *p* = 0.000; MSMD, trivial), and RT showed a very small effect size on decreasing the bench press muscle strength of women. The I^2^ statistic showed no statistical heterogeneity (I^2^, 0%, *p* = 0.933). RT showed a small effect on lean body mass in both men and women. RT interventions had a very small and insignificant effect on fat mass in both men and women.

#### 3.7.3. Effect of ET

Figure 15 presents the results of a meta-analysis of the effect of ET intervention on fat mass in men and women. The pooled SMD, which analyzed the results of the men using the Hedges’ g fixed-effects model, was −0.36 (95% CIs, −0.60 to −0.12; *p* = 0.003; MSMD, small), and ET intervention showed a small effect size on decreasing the fat mass in men. The I^2^ statistic showed no statistical heterogeneity (I^2^, 0%, *p* = 0.619). The pooled SMD, which analyzed the results of the women’s group, was −0.20 (95% CIs, −0.48 to 0.08; *p* = 0.168; MSMD, small), and ET showed a very small effect size on decreasing the fat mass of women. The I^2^ statistic showed no statistical heterogeneity (I^2^, 0%, *p* = 0.975). The intervention of ET showed a small effect size in fat mass reduction in the male group and a significant decrease; whereas, in the female group, ET showed a very small effect size in fat mass reduction, and no significant decrease was observed.

## 4. Discussion

The differences in the effects of ET, RT, and CT by sex and exercise type that was most effective for men and women were compared using a meta-analysis. To the best of our knowledge, this is the first study to include RCTs to verify sex differences in the effects of ET, RT, and CT interventions on muscle strength and body composition in men and women.

Regular physical activity, particularly RT, is widely recognized as an effective strategy to improve muscle strength and mass [53,54]. CT, combining RT and ET, may have higher muscle strength and muscular hypertrophy effects than ET or RT alone if performed with the appropriate exercise intensity and method [16,18]. After analyzing the results of 13 studies measuring lower body muscle strength using the leg press, both men and women showed large effect sizes for all exercise types. These results support several previous studies that examined changes in leg press strength in men and women according to exercise type [47,51]. For all exercise types, the effect size in the female group was larger than that in the male group. These results contradict those of several previous studies [47,55]. The male group included more studies than the female group and showed fewer discrepancies in the analysis results. In contrast, the female group included a small number of participants, and the error between the analysis results was relatively greater than that of the male group. The differences in effect size according to sex may be attributed to differences in weight, muscle fatigue, body composition distribution, and sex hormones; however, this is not yet clear, and additional studies are needed to directly compare men and women [56,57,58].

Comparing the effect size between exercise types on leg press strength within the male group, CT showed the largest effect size, followed by RT and ET. Therefore, the most effective type of exercise to increase leg press strength in men is CT. RT was found to be the most effective type of exercise to improve leg press strength in women. Although the ET group showed the smallest effect size for improving leg press muscle strength in both men and women, the ET group also demonstrated a large effect size. These results can be attributed to the use of a cyclometer in the ET group exercise protocol included in the analyses [16,31,41,47]. According to previous studies, muscle ET with cycle ergometers showed an increase in lower body strength similar to RT [51,59]. The most effective exercise types to increase leg press strength were CT in the male group and RT in the female group. These results can provide important information for the study of practical exercise programs.

After analyzing the results of nine studies measuring upper-body muscle strength using the bench press, the CT group showed the greatest effect size in both men and women. These results support those of previous studies in which CT intervention showed a greater increase in muscle strength than RT alone [34,41]. CT stimulates the secretion of testosterone, growth hormone, epinephrine, and norepinephrine. The interaction of these hormones significantly enhances muscle strength and physical performance [60]. CT effectively stimulates both type I and type II muscle fibers, promoting the development of a wide range of muscle fiber types. This stimulation not only enhances muscle strength but also significantly increases muscular endurance and power [61]. RT and CT had large effect sizes in both men and women. However, the ET intervention showed a medium effect size in the male group and a small effect size in the female group. Bench press strength is used as a representative measure of upper-body strength [46]. However, most ET protocols included in the analysis used a cycle ergometer and treadmill [31,33,41]. Such exercise methods may be more suitable for improving lower-body muscle function than upper-body muscle function, which could be a contributing factor to the smaller increase in bench press strength compared to leg press [51,59]. CT can be considered the most effective method to increase bench press strength in both men and women, and these results provide important information for the development of practical exercise programs aimed at increasing upper-body strength. However, to provide a more comprehensive understanding of the objective effects of combined CT interventions involving ET and RT on upper-body strength, additional studies utilizing protocols that incorporate upper-body exercises, circuit training, and interval training in addition to lower-body-focused ET protocols are needed.

The meta-analysis of 14 articles measuring lean body mass in this study revealed that CT interventions were the most effective in increasing lean body mass in both men and women, and RT showed a similar level of effectiveness as CT. However, the ET intervention had a very small effect size on lean body mass increase and was found to have no significant impact. These results contradict some previous studies [30,62]. If the ET protocol or exercise intensity of CT is inappropriate, it may limit muscle protein synthesis and increase muscle fatigue, potentially negatively affecting improvements in endurance and muscle mass [15,19,38]. Therefore, additional studies implementing a wider variety of exercise protocols are required to further demonstrate the effects of RT and CT on body mass enhancement. No sex-related differences were observed in the effect of exercise type on lean body mass. These findings contrast with those of previous studies indicating differences in lean body mass changes between men and women due to factors such as muscle protein metabolism, hormonal metabolism, and muscle fiber [63,64,65]. After additional research on lean body mass according to exercise type, it is necessary to conduct meta-analyses with larger sample sizes. A study by Aagaard et al., published in 2022, found no increase in lean fat after 14 weeks of RT intervention but found a significant increase in muscle strength [66]. Similarly, in this study, while the effect sizes of RT and CT interventions on fat mass reduction were small, significant effect sizes were observed for muscle strength increases. This seems to be the result of neural adaptation and may explain situations in which muscle strength is improved without increasing lean body mass [67]. In addition, to accurately compare muscle function and muscle mass improvement, it seems necessary to study muscle mass rather than lean body mass.

After conducting a meta-analysis of 16 articles measuring fat mass, it was found that the ET intervention was most effective in reducing fat mass in both men and women, although significant effect sizes were observed only in the male group. Therein, the intervention of ET and CT showed a small effect size on the reduction in fat mass and was found to have a statistically significant effect. These results support the findings of previous studies [30,68]. In contrast, in the female group, all exercise types showed very small effect sizes on fat mass reduction and were not statistically significant. ET and CT reduce the amount of fat in women [69,70]. In addition, the small effect may be attributed to the lack of study sample numbers, which was caused by the inclusion of only articles including ET, RT, and CT in the analysis. Therefore, additional studies are needed to investigate changes in fat mass in women depending on the type of exercise. The sex differences in the effects of ET and CT interventions on fat mass may be attributed to differences in energy metabolism and body weight between the sexes [56,71,72]. After the RT intervention, there was no significant decrease in fat mass in either the male or female groups. Because of the confusing results observed in many studies investigating the effects of RT intervention on fat mass, this remains difficult to prove [73,74,75]. In addition, the 17 studies included in the analysis had different periods of exercise intervention, ranging from 8 to 12 months. Changes in fat mass may vary depending on the duration of exercise; therefore, further studies are needed to analyze the effects of exercise type and duration.

This systematic review and meta-analysis had some limitations. First, our review may have reduced the number of articles included by limiting them to those published in English. Second, although only RCTs were included, heterogeneity existed among the analyzed studies, and the slightly asymmetrical funnel plot suggests the possibility of missing articles and the potential for underestimation of SMD. In addition, the risk of bias evaluation results showed a high risk for detection bias. Due to the nature of the study, even if the researcher does not notify the subject of what treatment he or she is receiving, the group proceeds separately into ET, RT, and CT. Therefore, if the average exercise period is more than 14 weeks, it is inevitable that subjects will be able to predict what treatment they are receiving. Therefore, additional RCTs that directly compare data between men and women are needed. Third, it is necessary to perform a subgroup analysis of these factors, because the analyzed variables may be affected by age, exercise period, and exercise volume. In particular, in the case of the exercise period, the average exercise period of the study dealing with men was 14 weeks, and the average exercise period of the study dealing with women was 16.9 weeks, which differed according to gender. Further research is needed to determine whether these differences in exercise periods have affected the results.

## 5. Conclusions

This systematic review suggests that the effect of exercise type on muscle strength and body composition may vary according to sex. CT showed the greatest effect size on increasing leg press strength in men, whereas RT and ET also showed large effect sizes. In contrast, in women, RT had the greatest effect on the increase in leg press strength. The exercise type that showed the greatest effect on increasing bench press strength was CT in both men and women. Therefore, CT can be considered the most effective exercise for increasing muscle strength in men. CT and RT showed small effect sizes for increasing lean body mass in both men and women. ET did not appear to have a significant effect on lean body mass. The effect size of ET appeared to be the largest among the different exercise types for reducing fat mass in both men and women, yet it was only small. In men, both ET and CT significantly reduced fat mass. However, no exercise type had a significant effect on fat mass reduction in women. The findings underscore the efficacy of CT in enhancing muscle strength, reducing fat mass, and improving lean body mass among male participants. Furthermore, these results provide useful information for the research and development of practical exercise programs based on sex. However, future research should explore CT’s effectiveness across diverse populations, including adolescents and older adults, to understand its broader applicability and benefits. While this study illuminates the advantages of CT, further research is essential to address methodological variations and demographic factors that may influence exercise outcomes. This includes exploring the impact of different exercise protocols and participant characteristics on the observed effects.

## Figures and Tables

**Figure 1 medicina-60-01186-f001:**
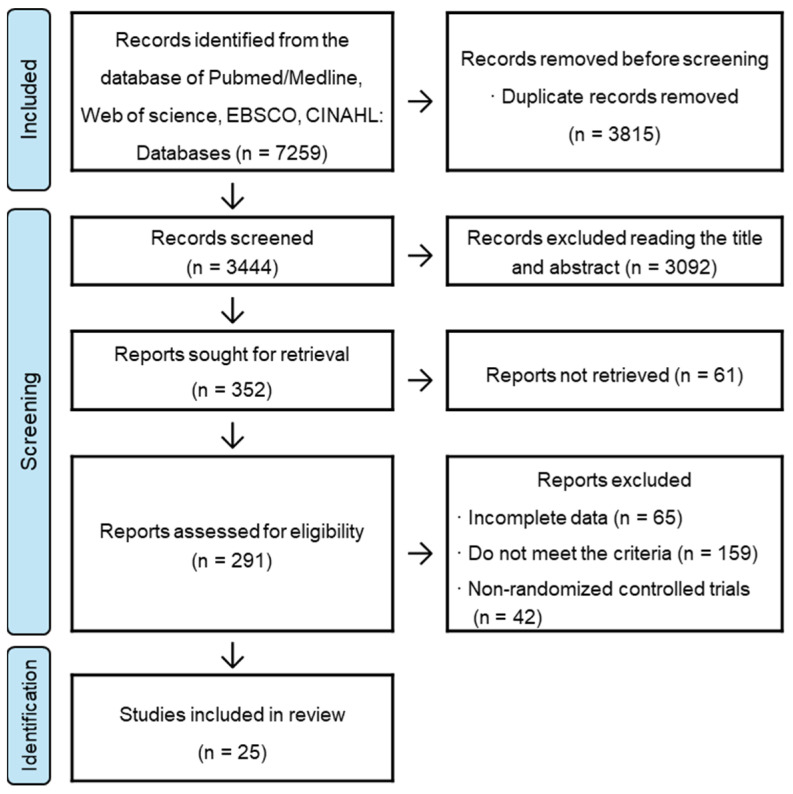
Flowchart of the search and study selection process.

**Figure 2 medicina-60-01186-f002:**
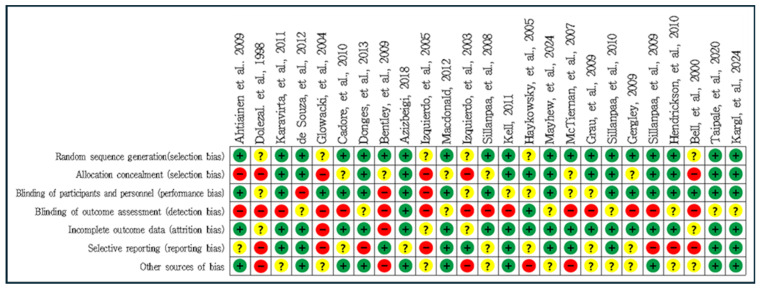
Risk of bias summary: judgments about each bias item for each study [16,18,30,31,32,33,34,35,36,37,38,39,40,41,42,43,44,45,46,47,48,49,50,51,52]. (+) indicates a low risk of bias, (?) indicates an unclear risk of bias, and (−) indicates a high risk of bias.

**Figure 3 medicina-60-01186-f003:**
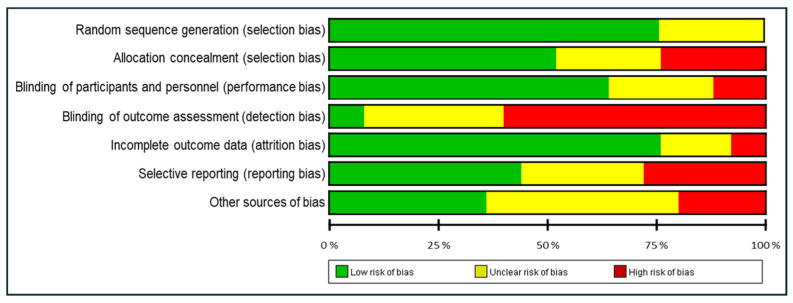
Risk of bias graph: review authors’ judgments about each risk of bias item, presented as percentages across all included studies.

**Figure 4 medicina-60-01186-f004:**
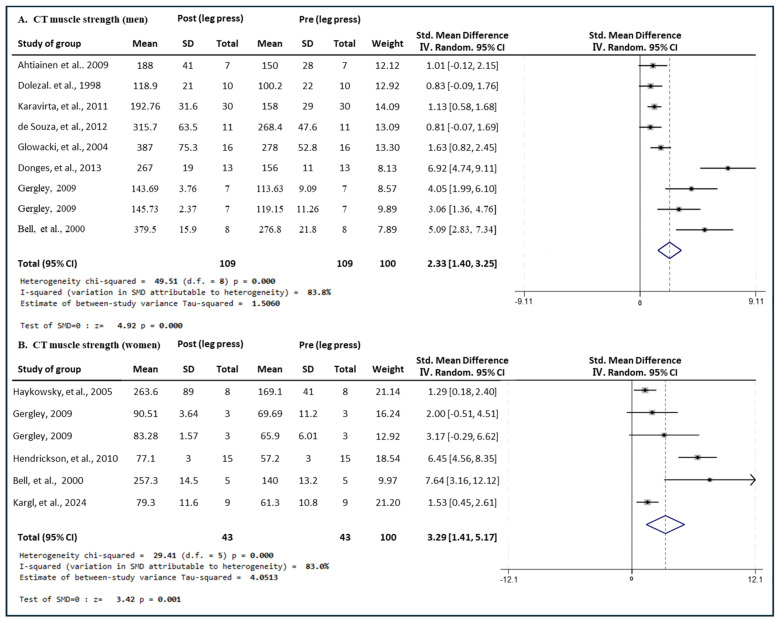
Meta−analysis results regarding changes in leg press 1 repetition maximum (1 RM) for men and women pre- and post-CT intervention. The forest plot shows the SMD with 95% confidence intervals (CIs) for the results of a leg press analysis of nine men’s groups (**A**) and six women’s groups (**B**). The diamond at the bottom shows the pooled SMD with the 95% CI for all studies following a meta-analysis. The horizontal lines represent the 95% CI of each study included in the meta−analysis. MD, mean difference; CI, confidence interval [16,30,31,32,33,35,41,46,47,49,51].

**Figure 5 medicina-60-01186-f005:**
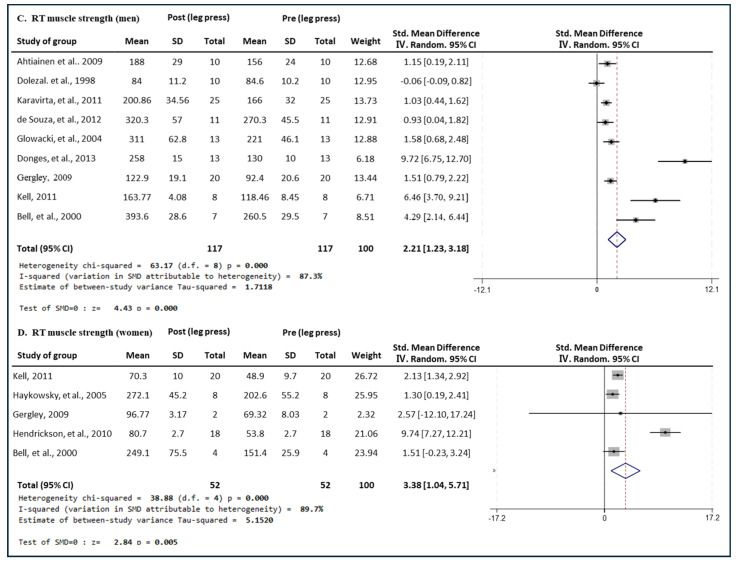
Meta−analysis results regarding changes in leg press 1 repetition maximum (1 RM) for men and women pre- and post-RT intervention. The forest plot shows the SMD with 95% confidence intervals (CIs) for the results of a leg press analysis of nine men’s groups (**C**) and five women’s groups (**D**). The diamond at the bottom shows the pooled SMD with the 95% CI for all studies following a meta-analysis. The horizontal lines represent the 95% CI of each study included in the meta−analysis. MD, mean difference; CI, confidence interval [16,30,31,32,33,35,40,41,46,47,51].

**Figure 6 medicina-60-01186-f006:**
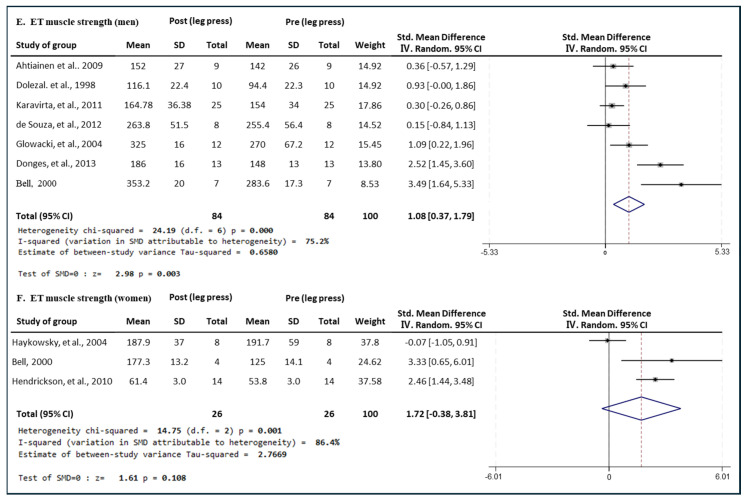
Meta−analysis results regarding changes in leg press 1 repetition maximum (1 RM) for men and women pre- and post-ET intervention. The forest plot shows the SMD with 95% confidence intervals (CIs) for the results of a leg press analysis of seven men’s groups (**E**) and three women’s groups (**F**). The diamond at the bottom shows the pooled SMD with the 95% CI for all studies following a meta−analysis. The horizontal lines represent the 95% CI of each study included in the meta−analysis. MD, mean difference; CI, confidence interval [16,30,31,32,33,35,41,46,47].

**Figure 7 medicina-60-01186-f007:**
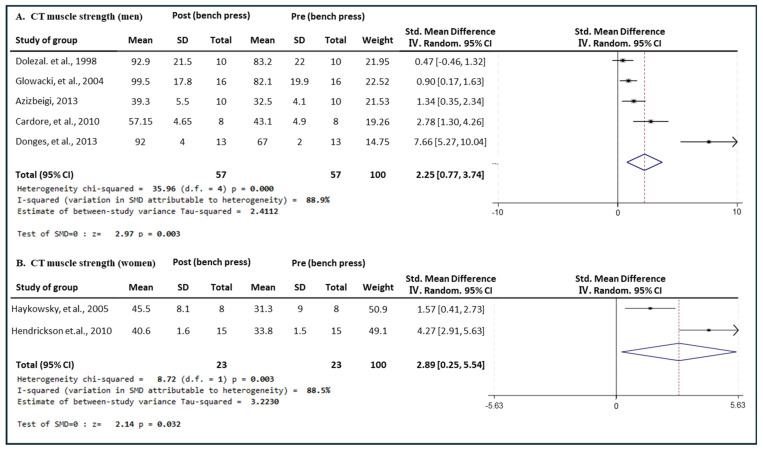
Meta−analysis results regarding changes in bench press 1 repetition maximum (1 RM) for men and women pre- and post-CT intervention. The forest plot shows the SMD with 95% confidence intervals (CIs) for the results of a bench press analysis of five men’s groups (**A**) and two women’s groups (**B**). The diamond at the bottom shows the pooled SMD with the 95% CI for all studies following a meta−analysis. The horizontal lines represent the 95% CI of each study included in the meta−analysis. MD, mean difference; CI, confidence interval [31,33,34,35,37,41,46].

**Figure 8 medicina-60-01186-f008:**
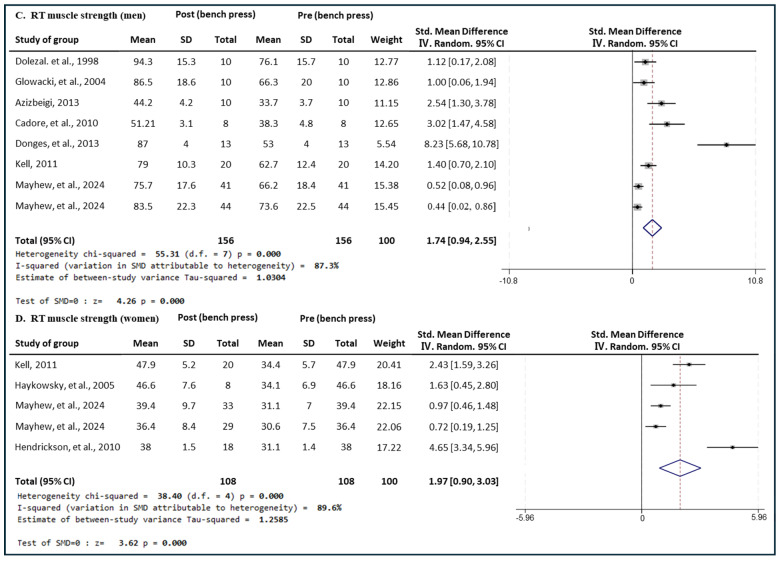
Meta−analysis results regarding changes in bench press 1 repetition maximum (1 RM) for men and women pre- and post-RT intervention. The forest plot shows the SMD with 95% confidence intervals (CIs) for the results of a bench press analysis of eight men’s groups (**C**) and five women’s groups (**D**). The diamond at the bottom shows the pooled SMD with the 95% CI for all studies following a meta−analysis. The horizontal lines represent the 95% CI of each study included in the meta−analysis. MD, mean difference; CI, confidence interval [31,33,34,35,37,40,41,42,46].

**Figure 9 medicina-60-01186-f009:**
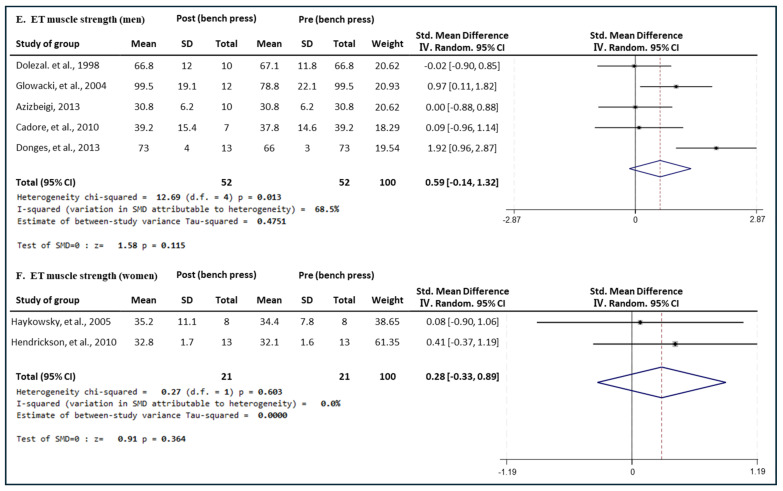
Meta−analysis results regarding changes in bench press 1 repetition maximum (1 RM) for men and women pre- and post-ET intervention. The forest plot shows the SMD with 95% confidence intervals (CIs) for the results of a bench press analysis of five men’s groups (**E**) and two women’s groups (**F**). The diamond at the bottom shows the pooled SMD with the 95% CI for all studies following a meta−analysis. The horizontal lines represent the 95% CI of each study included in the meta−analysis. MD, mean difference; CI, confidence interval [31,33,34,35,37,41,46].

**Figure 10 medicina-60-01186-f010:**
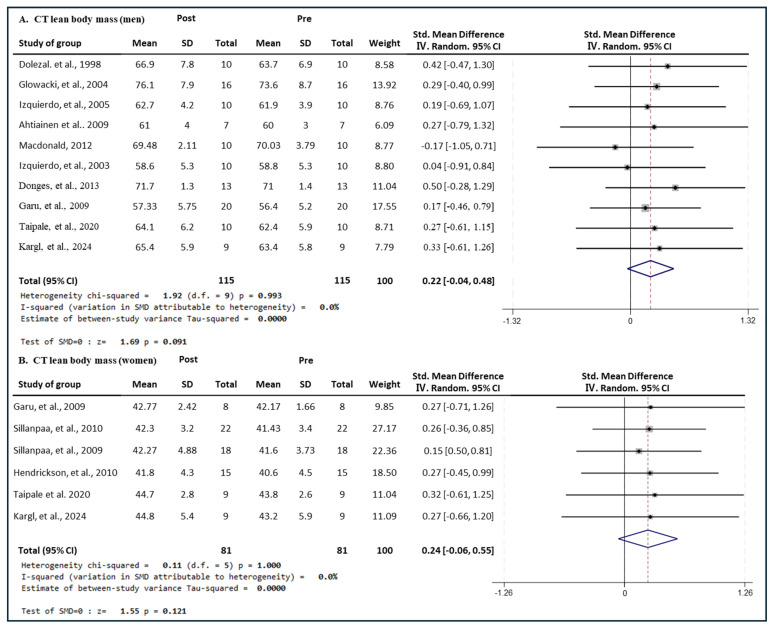
Meta−analysis results regarding changes in lean body mass (kg) for men and women pre- and post-CT intervention. The forest plot shows the SMD with 95% confidence intervals (CIs) for the results of a lean body mass analysis of ten men’s groups (**A**) and six women’s groups (**B**). The diamond at the bottom shows the pooled SMD with the 95% CI for all studies following a meta−analysis. The horizontal lines represent the 95% CI of each study included in the meta−analysis. MD, mean difference; CI, confidence interval [18,30,31,33,35,38,39,44,45,46,48,49].

**Figure 11 medicina-60-01186-f011:**
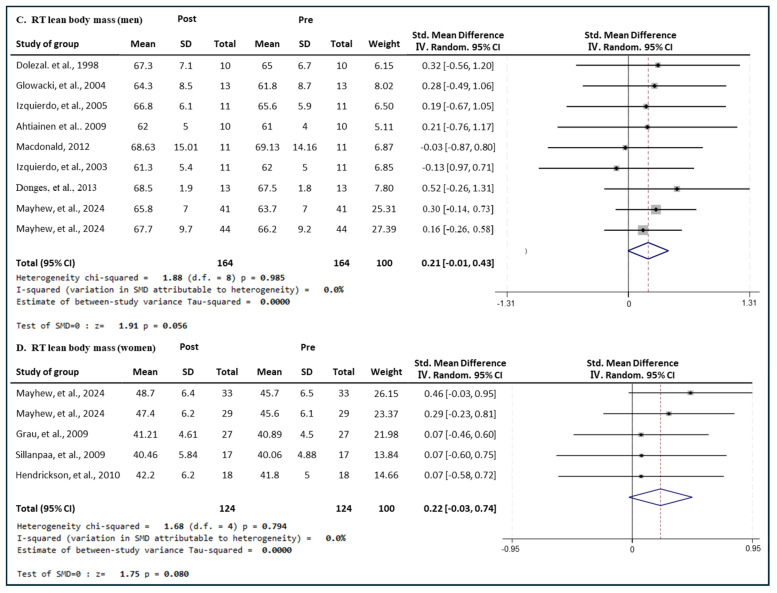
Meta−analysis results regarding changes in lean body mass (kg) for men and women pre- and post-RT intervention. The forest plot shows the SMD with 95% confidence intervals (CIs) for the results of a lean body mass analysis of nine men’s groups (**C**) and five women’s groups (**D**). The diamond at the bottom shows the pooled SMD with the 95% CI for all studies following a meta−analysis. The horizontal lines represent the 95% CI of each study included in the meta−analysis. MD, mean difference; CI, confidence interval [18,30,31,33,35,38,39,42,44,45,46].

**Figure 12 medicina-60-01186-f012:**
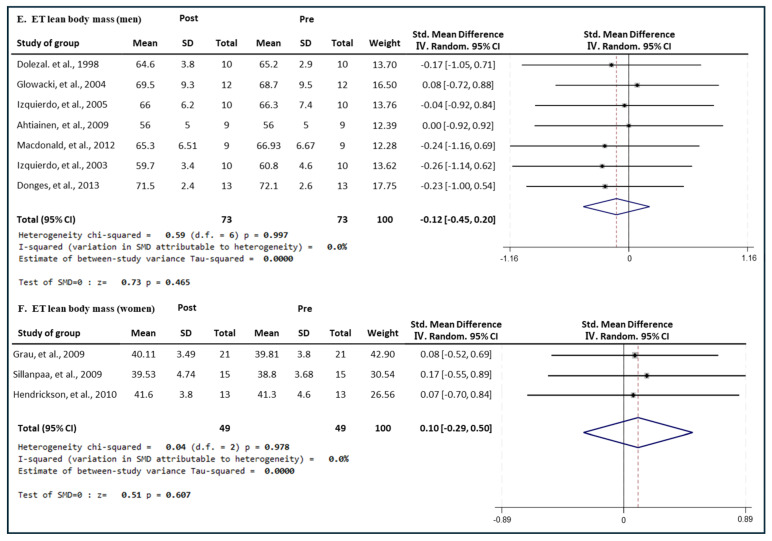
Meta−analysis results regarding changes in lean body mass (kg) for men and women pre- and post-ET intervention. The forest plot shows the SMD with 95% confidence intervals (CIs) for the results of a lean body mass analysis of seven men’s groups (**E**) and three women’s groups (**F**). The diamond at the bottom shows the pooled SMD with the 95% CI for all studies following a meta−analysis. The horizontal lines represent the 95% CI of each study included in the meta−analysis. MD, mean difference; CI, confidence interval [18,30,31,33,35,38,39,44,45,46].

**Figure 13 medicina-60-01186-f013:**
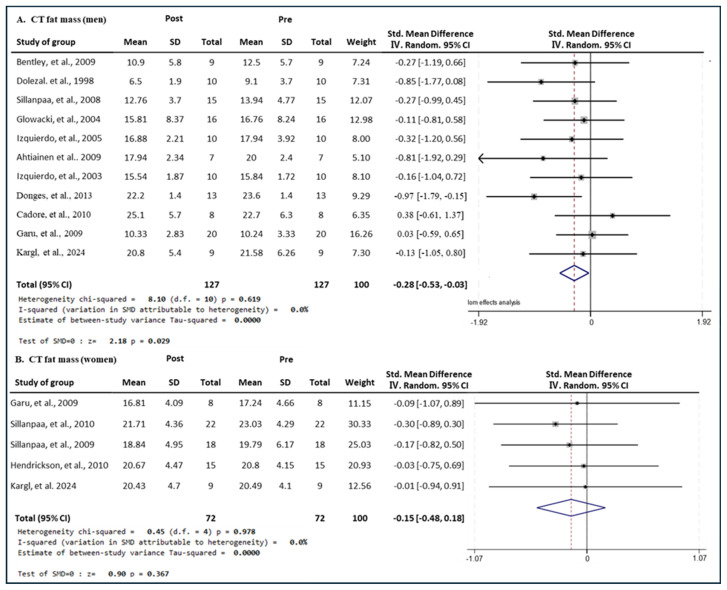
Meta−analysis results regarding changes in lean fat mass (kg) for men and women pre- and post-CT intervention. The forest plot shows the SMD with 95% confidence intervals (CIs) for the results of a fat mass analysis of eleven men’s groups (**A**) and five women’s groups (**B**). The diamond at the bottom shows the pooled SMD with the 95% CI for all studies following a meta−analysis. The horizontal lines represent the 95% CI of each study included in the meta−analysis. MD, mean difference; CI, confidence interval [18,30,31,33,34,35,36,39,44,45,46,49,52].

**Figure 14 medicina-60-01186-f014:**
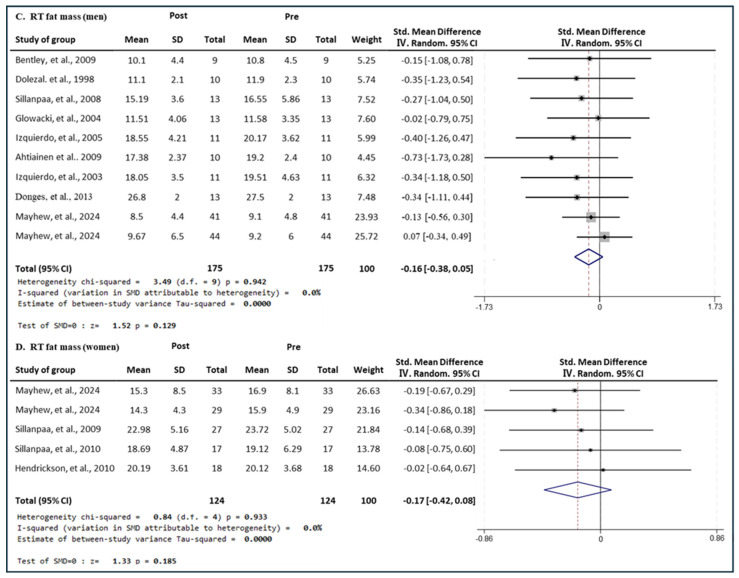
Meta−analysis results regarding changes in lean fat mass (kg) for men and women pre- and post-RT intervention. The forest plot shows the SMD with 95% confidence intervals (CIs) for the results of a fat mass analysis of nine men’s groups (**C**) and four women’s groups (**D**). The diamond at the bottom shows the pooled SMD with the 95% CI for all studies following a meta−analysis. The horizontal lines represent the 95% CI of each study included in the meta−analysis. MD, mean difference; CI, confidence interval [18,30,31,33,35,36,39,42,45,46,50,52].

**Figure 15 medicina-60-01186-f015:**
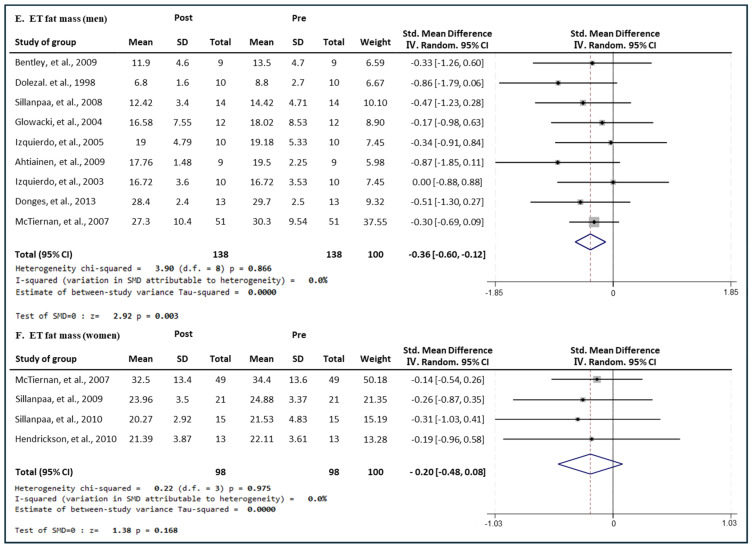
Meta−analysis results regarding changes in lean fat mass (kg) for men and women pre- and post-ET intervention. The forest plot shows the SMD with 95% confidence intervals (CIs) for the results of a fat mass analysis of eleven men’s groups (**E**) and five women’s groups (**F**). The diamond at the bottom shows the pooled SMD with the 95% CI for all studies following a meta−analysis. The horizontal lines represent the 95% CI of each study included in the meta−analysis. MD, mean difference; CI, confidence interval [18,30,31,33,35,39,43,45,46,47,50,52].

**Table 1 medicina-60-01186-t001:** Summary of study characteristics of included studies.

Study	Study Population	Groups	Sex	Age	N	Exercise Intervention		Outcome
						Exercise Program	Frequency × Duration	
Ahtiainen et al., 2009 [30]	Middle aged men	ET	Men	58	9	Used to bicycle ergometer and HR, 45–90 min, progressive overload	2 days/week × 21 weeks	LP, LBM, FM
		RT		61	10	1 RM 40–90%, 6–30 reps/set, progressive overload		
		CT		64	7	ET + RT concurrent training	4 days/week × 21 weeks	
Dolezal et al., 1998 [31]	Young aged men	ET	Men	20.1	10	Used to treadmill, intensity of HRmax 65~85%. Progressive overload	3 days/week × 10 weeks	LP, BP, LBM, FM
		RT			10	4–15 reps/set, 3 sets, progressive overload		
		CT			10	Half of ET + RT		
Karavirta et al., 2011 [16]	Middle aged men	ET	Men	54	25	Used to cycle ergometer and aerobic thresholds (30–60 min)	2 days/week × 21 weeks	LP
		RT		56	25	1 RM 40–85% 5~20 reps/set, 2–4 sets, progressive overload		
		CT		56	30	ET + RT concurrent training	4 days/week × 21 weeks	
de Souza et al., 2012 [32]	Young aged men	ET	Men	24	8	Used to high intensity interval training on treadmill, VO2max 80–100%	2 days/week × 8 weeks	LP
		RT		25.9	11	6–12 RM reps/set, lower body muscle		
		CT		22.5	11	ET + RT concurrent training		
Glowacki et al., 2004 [33]	Young aged men	ET	Men	25	12	Used to treadmill, HR reserve 65~80%, progressive overload	2–3 days/week × 12 weeks	LP, BP, LBM, FM
		RT		23	13	1 RM 75–85% progressive overload		
		CT		22	16	RT (3 days/week) + ET (2 days/week) × 6 weeks, ET (3 days/week) + RT (2 days/week) × 6 weeks	5 days/week × 12 weeks	
Cadore et al., 2010 [34]	Older men	ET	Men	64.4	7	Used to cycle ergometer, HRmax 80% 20 min (~10 week), HRmax 100% 6 sets × 4 min (11–12 week)	3 days/week × 8 weeks	BP, FM
		RT		64.0	8	6–20 RM reps/set, progressive overload		
		CT		66.8	8	ET + RT concurrent training		
Donges et al., 2013 [35]	Middle aged men	ET	Men	45.4	13	Used to cycle ergometer, HRmax 75–80% (40~60 min)	3 days/week × 12 weeks	LP, BP, LBM, FM
		RT		51.7	13	1 RM 75–80%, 8–10 reps/set, 3–4 sets		
		CT		46.2	13	Half of ET + RT concurrent training		
Bentley et al., 2009 [36]	Young aged men	ET	Men	24.8	9	HRmax 65% Progressive overload	3 days/week × 8 weeks	FM
		RT		25.4	9	1 RM 50%, 10 reps, 2 set, progressive overload		
		CT		24.4	9	Half of ET + RT concurrent		
Azizbeigi, 2018 [37]	Young aged men	ET	Men	21.1	10	HRmax 50–85%, progressive overload	3 days/week × 8 weeks	BP
		RT		21.2	10	1 RM 50–85%, progressive overload		
		CT		22.8	10	Alternatively every week, first week: RT, second week: ET		
Izquierdo et al., 2005 [18]	Middle aged men	ET	Men	42.3	10	Used to cycle ergometer, HRmax 70–90%, 30–40 min	2 days/week × 12 weeks	LBM, FM
		RT		43.5	11	1 RM 50–70%, 5~10 reps/set, 3–4 sets, progressive overload		
		CT		41.8	10	ET (1 day/week) + RT (1 day/week)		
Macdonald, 2012 [38]	Young aged men	ET	Men	20.56	9	Plyometric training, 3–7 reps/set 3 sets	2 days/week × 9 weeks	LBM
		RT		22	11	Training day 1 (1 RM 75–90%, 3–6 reps/set, 3 sets) Training day 2 (1 RM 45–67%, 3–6 reps/set, 3 sets)		
		CT		22.50	10	ET + RT concurrent training		
Izquierdo et al., 2003 [39]	Older men	ET	Men	68.2	10	Used to cycle ergometer, HRmax 55–85%, 30–40 min	2 days/week × 16 weeks	LBM, FM
		RT		64.8	11	1 RM 50–70%, 10–15 reps/set, 3–4 sets (1~8 week) 1 RM 70–80%, 5–6 reps/set, 3–5 sets (9–16 week)		
		CT		66.4	10	ET (1 day/week) + RT (1 day/week) concurrent training		
Sillanpaa et al., 2009 [50]	Middle aged women	ET	Women	51.7	15	1–7 weeks: 30 min cycling; 8–14 weeks: 45–60 min cycling; 15–21 weeks: 60–90 min cycling	2 days/week × 21 weeks	LBM, FM
		RT		50.8	17	1–7 weeks: 1 RM 40–60%, 15–20 reps 8–14 weeks: 1 RM 60–80%, 10–12 reps 15–21 weeks: 1 RM 70–90%, 6–8 reps		
		CT		48.9	18	ET + RT		
Hendrickson et al., 2010 [46]	Recreation-ally active women	ET	Women	21	13	HRmax 70–85%, 20–30 min, 400, 800, 1200, 1600 m jogging	3 days/week × 12 weeks	LP, BP LBM, FM
		RT		21	18	1–2 weeks: Familiarization period 3–6 weeks: Light intensity: 12 RM (rest 90 sec), Moderate: 8–10 RM (rest 120 s), Heavy: 6–8 RM (rest 120 s) × 3 sets (all intensity) 8–11 weeks: Light intensity: 12 RM (rest 90 sec), Moderate: 6–8 RM (rest 150 s), Heavy: 3–5 RM (rest 180 s) × 3 sets (all intensity)		
		CT		20	15	First week: ET (3 day/week)/Next week: RT (3 day/week) (Alternate week)		
Taipale et al., 2020 [48]	Recreation -ally trained men and women	CT	Men	32.6	10	2 RT + 2 ET, Progressive overload RT: 1 RM 50–85% (Main exercise, 2 days/week), Plyometric A (Once per week), Plyometric B (Once per week) (After main exercise, core and upper body exercise) ET: HIIT HRmax 70–90% 4 min × 4 sets + HRmax 60–70% 4 min × 3 sets + 100 m running all-out	4 days/week × 10 weeks	LBM
			Women	31.3	9			
Kargl et al., 2024 [49]	Healthy young men and women	CT	Men	27.3	9	RT General Physical Preparedness: 2 weeks, 3 sets, 10 reps, 1 RM 64–72% Preparation for Peak Force Production: 1 week, 3–4 sets, 5–6 reps, 1 RM 72–80% Peak Force Development: 3 weeks, 3–5 sets, 3 reps, 1 RM 79–88% Rate of Force Development: 3 weeks, 3–4 sets, 2–3 reps, 1 RM 81–90% ET HRmax 70–85%, 60–90 min, runs and sprints	12 weeks	LP, LBM, FM
			Women	27.4	9			
Bell et al., 2000 [47]	Men and women	ET	Men	20.75	7	Cycle: Increasing every 4 min per 4 weeks (30–42 min) (2 days/week), Cycle interval: Increasing every 1 sets per 4 weeks (4–7 sets) (1 day/week)	3 days/week × 12 weeks	LP
		RT		20	7	Upper and lower body exercise Increasing every 4%/3 week (1 RM 72–84%), 4–12 reps, 2–6 sets		
		CT		19.42	8	ET + RT alternating days (RT: 3 days/week; ET: 3 days/week)	6 days/week × 12 weeks	
		ET	Women	20.5	4	Same as the exercise program in the men group.	3 days/week × 12 weeks	
		RT		21	4			
		CT		20.33	5		6 days/week × 12 weeks	
Sillanpaa et al., 2008 [52]	Middle aged men	ET	Men	54.1	14	Used to cycle ergometer, progressive overload (30–90 min)	2 days/week × 21 weeks	FM
		RT		54.6	13	1 RM 40–90%, progressive overload		
		CT		56.3	15	ET + RT concurrent training (RT: 2 days/week; ET: 2 days/week)	4 days/week × 21 weeks	
Kell, 2011 [40]	Young aged men and women	RT	Men	22.7	20	4 weeks mesocycle Week 1–2: Whole-body workout session/moderate volume (3–4 set; 324–640 reps), lower intensity (55–57% 1 RM) Week 3: adding 4 new exercises (split routine 4 days/week) Week 4: week 4 was a testing and recovery week	3–4 days/week × 12 weeks	LP, BP
		RT	Women	22.5	20			
Haykowsky et al., 2005 [41]	Older women	ET	Women	66	8	Cycle exercise at an intensity between 60% and 80% of HR/progressive intensity (2.5 min every week up to a maximal duration of 42.5 min	3 days/week × 12 weeks	LP, BP
		RT		70	8	Upper and lower extremity/2 sets of 10 reps/progressive overload (2.5% every week) (progressively increased to 75% of 1 RM)		
		CT		68	8	ET + RT concurrent training		
Mayhew et al., 2024 [42]	Young aged men and women	RT65%	Men	19.5	41	1 RM 65% BP and Squat 3 sets × 10–12 reps (5 weeks)/3 sets × 6–8 reps (4 weeks)/3 × 3–5 RM (3 weeks)	3 days/week × 12 weeks	LP
			Women	19.2	33			
		RT90%	Men	19.5	44	1 RM 90% BP and squat 3 set × 10–12 reps (5 weeks)/3 × 6–8 RM (4 weeks)/3 × 3–5 RM (3 weeks)		
			Women	19	29			
McTiernan et al., 2007 [43]	Men and women	ET	Men	56.2	51	60 min session, moderate-to-vigorous aerobic exercise gradually achieved over the first 12 weeks.	12 months, 6 days/week	FM
		ET	Women	54.4	49			
Grau et al., 2009 [44]	Young aged men and women	CT	Men	23.13	20	RT + plyometric jumps: 1 RM 50%, 1 set × 12 reps/1 RM 70% 1–3 set × 6–10 reps/1 RM 90% 1–2 set × 2–3 reps + Drop jump 4–11 set × 5/hurdles 4–11 set × 5	3 days/week × 9 weeks	LBM, FM
			Women	22.3	8			
Gergley et al., 2009 [51]	Healthy young men, women	RT	Men	20.75	8	1–3 weeks: 3 × 12 RM, (Leg extension/flexion), (Leg press); 4–6 weeks: 3 × 10 RM, (Leg extension/flexion), (Leg press); 7–9 weeks: 3 × 8 RM, rest 150 sec (Leg extension/flexion)/3 × 8 RM, rest 180 sec (Leg press)	2 days/week × 9 weeks	LP
		CT 1		20	7	RT + 1–3 weeks: HRmax 65%, 20 min (Cycle) 4–6 weeks: HRmax 65% 30 min (Cycle) 7–9 weeks: HRmax 65% 40 min (Cycle)		
		CT 2		19.42	7	The same intensity as CT 1 (Treadmill)		
		RT	Women	20.5	2	The same protocol as men’s groups		
		CT 1		21	3			
		CT 2		20.33	3			
Sillanpaa et al., 2010 [45]	Middle aged women	ET	Women	53	21	Progressive overload by cycling (1 cycle: 30 min, 2 cycle: 45 min, 3 cycle: 50–60 min	2 days/week 21 weeks	LBM, FM,
		RT		52	27	10 rm, 3–4 sets: cycle 1 (1–7 week): 1 RM 40–60%, cycle 2 (8–14 week): 1 RM 60–80%, cycle 3 (15–21 week): 1 RM 70–90%		
		CT		51	22	ET + RT		

ET, endurance training; RT, resistance training; CT, concurrent training; N, number; HR, heart rate; RM, repetition maximum; LP, leg press; BP, bench press; LBM, lean body mass; FM, fat mass; VO_2max,_ maximal oxygen consumption; rep, repetition; min, minute.

## Data Availability

Data available in a publicly accessible repository that does not issue DOIs for publicly available datasets were analyzed in this study. This data can be found here (https://figshare.com/articles/figure/Effects_of_Exercise_Types_on_Muscle_Strength_and_Body_Composition_in_Men_and_Women_A_Systematic_Review_and_Meta-analysis/26083729, accessed on 22 June 2024).

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
