# Peer review of "Effects of Exercise Type on Muscle Strength and Body Composition in Men and Women: A Systematic Review and Meta-Analysis"

_medicina, 2024, doi:10.3390/medicina60071186_

Round 1

Reviewer 1 Report

Comments and Suggestions for Authors

Respected Authors 

My comments on your study 

-The conclusion if written in bulleted points highlighting the final inferences of the authors will be most useful and clear to the readers 

- The final paragraph of the conclusion section mainly mention about limitation of the study , hence this can be written in a separate section with heading limitation of the study .

- There has to be a mention of average duration of exercise period in the discussion or conclusion section based on all the studies which may be of beneficial to either sexes .

Thank you 

-

Comments on the Quality of English Language

No editing required for English language 

Reviewer 2 Report

Comments and Suggestions for Authors

1. Line 48-61: While it includes relevant information on sarcopenia and obesity, it goes deeper too deeply into public health issues, which detracts from the specific focus on exercise types and sex differences. Please consider trimming down the general health information and focus more on the gap this study aims to fill.

2. Line 126: Please provide more rationale for only including studies in English as this may introduce bias.

3. Figure 3: There is a high risk for detection bias as per the figure (especially with few studies, please provide the implications of this or discuss this risk in detail in the discussion).

4. Table 1: The table is overwhelming to the reviewer, please highlight the common themes and characteristics of all the included studies by adding more content to paragraph 3.3 so that the reviewer/ reading can understand it better.

5. I2 statistic of some the studies are very high which needs discussion in terms of interpretation/ implications in the discussion section which is missing currently.

6. All figures: please keep the same x limit for both men and women – this will aid easy comparison. For example, in figure 4, keep the men’s x limit at -12.1 and 12.1.

7. It will be nice to have a list of abbreviations on the paper.

8. Line 554: Please make sure the WHO report is up to date.

Reviewer 3 Report

Comments and Suggestions for Authors
